# Formation Control of Swarming Vessels Using a Virtual Matrix Approach and ISOT Guidance Algorithm

**Su-Rim Kim** [1] , **Hyun-Jae Jo** [1] , **Jung-Hyeon Kim** [1] **and Jong-Yong Park** [2],*

1   Department of Marine Design Convergence Engineering, Pukyong National University, Busan 48513, Korea; fla9898@pukyong.ac.kr (S.-R.K.); hyunjae9301@pukyong.ac.kr (H.-J.J.); hyeony96@pukyong.ac.kr (J.-H.K.)
2   Department of Naval Architecture and Marine System Engineering, Pukyong National University, Busan 48513, Korea
*   Correspondence: jongyong.park@pknu.ac.kr; Tel.: +82-51-629-6619

**Abstract:** The formation control for the effective operation of multiple vessels is discussed. First, a virtual matrix approach is proposed to improve the formation robustness and transform performance during swarm operations, which is created based on the virtual leader vessel location, and agents composing the formation follow cells in the matrix to maintain formation. This approach is affected by the virtual leader vessel location. The virtual leader vessel location is defined by two cases: matrix center and geometric center; furthermore, robustness and efficiency comparison simulations are performed. The simulation results show that in most formations, the geometric center is better in terms of efficiency and robustness. Second, the isosceles triangle guidance algorithm is proposed to improve the "go-back behavior" of certain agents during excessive maneuvering. Through a waypoint-following simulation, the algorithm is confirmed to be superior to the line-of-sight guidance algorithm. The swarm simulation on the virtual map verifies the performance of the proposed formation control and guidance algorithm.

**Keywords:** autonomous surface vessel (ASV); swarm operation; formation control; virtual matrix approach; virtual leader vessel; isosceles triangle (ISOT)



## 1. Introduction

As the application of unmanned systems in maritime missions has received much attention, discussions regarding autonomous surface vessels (ASVs) have dominated research in recent years, such as auto-berthing and unberthing systems [1] and collision avoidance algorithms [2,3]. Based on previous research, ASVs have been extensively applied to maritime missions. The AUTOSHIP project uses full autonomous navigation, self-diagnostic prognostics, and operation scheduling for autonomous multimodal transport to relieve road congestion and associated pollution [4,5]. Further, YARA Birkeland, equipped with a fully autonomous mooring system, is aimed at operating berthing and unberthing without human intervention [6].

As ASVs are applied to actual situations and their maritime missions are gaining importance, a high standard is required in terms of time and accuracy. In the case of a single ASV, there is a possibility that the missions are limited owing to certain capability limitations, such as short duration and failures of sensors or actuators. Therefore, swarm operations wherein multiple vessels cooperate to perform one mission are studied to increase efficiency. A multi-mobile robot formation control system was applied to various fields by mapping the underwater environment [7–10].

The formation control algorithm is the fundamental system for maintaining and transforming the formation of a swarm operation, and there has been a growing body of research in this regard. According to prior studies, the leader–follower and virtual structure approaches are widely accepted for formation control. The leader–follower approach sets one or several vessels among all entities constituting a formation as a leader and uses the

relative geometrical relationship between the leader and follower, such as the heading angle and distance, to define the followers' position [11,12]. A variety of solutions have been proposed to increase the stability of the leader–follower approach. Various control methods, such as sliding mode control [13], fuzzy logic control [14], linear quadratic regulator control [15], and feedback linearization control [16] were applied to the relative heading angle and distance. A guided formation control scheme was developed using a modular design procedure inspired by concepts from integrator backstepping and cascade theory [17]. Further, there have been many attempts to overcome the weak connection and environmental disturbance when approaches are applied to real environments. In [18–21], a decentralized leader–follower approach is proposed wherein each agent measures local information about the surrounding vessels to provide reliable control when communication between agents is limited or even unavailable. The adaptive formation control algorithm using radial-basis-function neural networks (RBF NNs) and the minimal learning parameter (MLP) have been presented to compensate for model uncertainty perturbations and environmental disturbances [22].

In contrast, in the virtual structure approach, the entire formation is treated as a single rigid body and the motion of each agent is derived from the trajectory of a corresponding point on the structure [23,24]. Prior studies have focused on algorithms that apply formation feedback to increase the formation accuracy [25–30]. The concept of a decentralized structure was introduced, which causes the mission to continue to perform even if a problem occurs in one entity [31]. In [32], a simulation is provided to verify the combined concept of feedback linearization control and a decentralized structure.

In this study, we propose the virtual matrix approach inspired by the virtual structure approach concepts. Isosceles triangle guidance (ISOT) is suggested as a suitable algorithm for the virtual matrix approach for the formation of multiple ASVs. The virtual matrix approach creates a virtual matrix to give rise to a formation and works as a rigid body in the virtual structure approach. The virtual matrix is calculated by the virtual leader vessel location and the distance between rows and columns. The agent maintains the formation by following the designated cells in the matrix that follows the line-of-sight (LOS) guidance algorithm. As opposed to the virtual structure approach, which requires a specific distance and angle of each agent from the reference point, the virtual matrix approach allows simple control of formation maintenance and transformation by assigning only the cell number to the agent. The virtual leader vessel location is defined as the matrix center (MC) and geometric center (GC), which affect the performance of the formation control. Further, a waypoint-following simulation was performed to compare the robustness and efficiency of this approach. Certain agents following the LOS guidance algorithm exhibited inefficient behavior during excessive maneuvering. The ISOT guidance algorithm was proposed to improve this movement. A waypoint-following simulation was performed to verify the performance of the ISOT guidance algorithm, and the results were compared with those of the LOS guidance algorithm. The swarm simulation on the virtual map was provided to verify the maintenance and transform performance of the proposed formation control and guidance algorithm.

The remainder of this paper is organized as follows. Section 2 introduces the ASV dynamic model used in this article. In Section 3, the virtual matrix approach is described and a performance comparison according to the virtual leader vessel location is presented. Section 4 contains a detailed description of the ISOT guidance algorithm. Here, the ISOT and LOS performance comparison results are presented. In Section 5, swarm simulation using the virtual matrix approach and ISOT guidance algorithm is performed. Finally, Section 6 concludes this article and discusses the prospects for further work.

## 2. ASV Dynamic Model

In this study, a wave adaptive modular vessel (WAM-V) was used for the swarm operation simulations [33]. The WAM-V was propelled using the thrusters installed to the left and right sides in the form of a catamaran, which can turn owing to the difference in

thrust between the left and right sides, without steering. The specifications of the ASV are presented in Table 1.

**Table 1.** Principal specifications of the ASV.

| Parameter | Value |
|:---:|:---:|
| Length | 4.88 m |
| Beam | 2.50 m |
| Draft | 0.30 m |
| Mass | 151 kg |
| Moment of Inertia | 257 kg × m |

The coordinate system used to express the horizontal motion of the ASV, such as surge, sway, and yaw, is shown in Figure 1 [34]; $x_o$ and $y_o$ denote the earth-centered fixed-coordinate system, $\psi$ denotes the heading angle of the ASV, and $u$ and $v$ denote the surge and sway velocities, respectively.

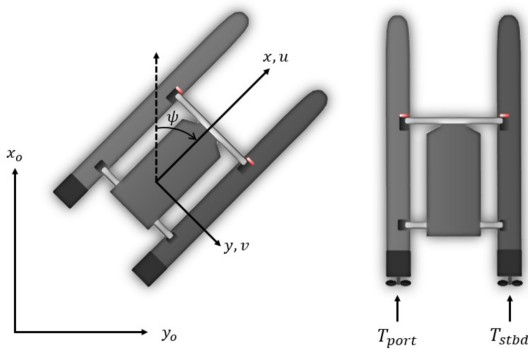

**Figure 1.** Coordinate system of the ASV.

The position vector of the ASV and speed vector are defined in Equations (1) and (2), respectively. They have the relationship shown in Equation (3). Further, $R(\eta)$ in Equation (4) is a rotation matrix that converts the body-centered fixed-coordinate system to an earth-centered fixed-coordinate system.

$$\eta = [x\ y\ \psi]^T \tag{1}$$

$$\nu = [u\ v\ r]^T \tag{2}$$

$$\eta = \dot{R}(\eta)\nu \tag{3}$$

$$R(\eta) = \begin{bmatrix} \cos(\psi) & -\sin(\psi) & 0 \\ \sin(\psi) & \cos(\psi) & 0 \\ 0 & 0 & 1 \end{bmatrix} \tag{4}$$

The thruster of the ASV is fixed to the hull and has a propulsive force in the longitudinal direction of the hull; thus, it has a control force and moment, as shown in Equation (5). $T_{port}$ and $T_{stbd}$ are the thrust forces of the port and starboard side thrusters, respectively, and $B$ is the beam of the ASV described in Table 1.

$$f = \begin{bmatrix} \tau_X \\ \tau_Y \\ \tau_Z \end{bmatrix} = \begin{bmatrix} T_{port} + T_{stbd} \\ 0 \\ (T_{port} - T_{stbd}) \times B/2 \end{bmatrix} \tag{5}$$

The thrust according to the thruster's RPM $\delta n$ of the ASV is calculated using Equations (6) and (7) [33]. Equation (6) expresses the thrust during the forward RPM, and Equation (7) expresses the thrust during the reverse RPM.

$$T = 3.54 \times 10^{-5} \delta n^2 + 0.084 \delta n - 3.798 \tag{6}$$

$$T = -1.189 \times 10^{-5} \delta n^2 + 0.071 \delta n + 4.331 \tag{7}$$

The linearized dynamic model is used in Equation (8) [33], and Equations (9)–(11) are the unknown parameters of the linearized dynamic model.

$$\dot{v} = Av + Bf + C \tag{8}$$

$$A = \begin{bmatrix} -1.0191 & 0 & 0 \\ 0 & 0.0161 & -0.0052 \\ 0 & 8.2861 & -0.9860 \end{bmatrix} \tag{9}$$

$$B = \begin{bmatrix} 0.0028 & 0 & 0 \\ 0 & 0 & 0.0002 \\ 0 & 0 & 0.0307 \end{bmatrix} \tag{10}$$

$$C = \begin{bmatrix} 0.6836 \\ 0.0068 \\ 1.3276 \end{bmatrix} \tag{11}$$

In this study, ASV waypoint tracking was performed using LOS, as shown in Equation (12); $\psi_d$ is the desired heading angle, $x$ and $y$ are the positions of the ASV, and $x_w$ and $y_w$ are the positions of the waypoints.

$$\psi_d = \tan^{-1} \frac{(y_w - y_s)}{(x_w - x_s)} \tag{12}$$

To control the heading angle and velocity of the ASV, proportional integral derivative (PID) control from Equations (13)–(16) was used.

$$\delta n_\psi = k_{p,\psi} e_\psi + k_{i,\psi} \int e_\psi dt + k_{d,\psi} \dot{e}_\psi \tag{13}$$

$$e_\psi = \psi_d - \psi \tag{14}$$

$$\delta n_V = k_{p,V} e_V + k_{i,V} \int e_V dt + k_{d,V} \dot{e}_V \tag{15}$$

$$e_V = \sqrt{(x_w - x_s)^2 + (y_w - y_s)^2} \tag{16}$$

where $\delta n_\psi$ and $\delta n_V$ are the heading angle and velocity PID control values, respectively; $e_\psi$ and $e_V$ are the heading angle and distance errors, respectively; $k_{p,\psi}$, $k_{i,\psi}$, and $k_{d,\psi}$ are the P, I, and D control gains of the heading angle PID control, respectively; and $k_{pV}$, $k_{i,V}$, $k_{d,V}$ denote the P, I, and D control gains of the velocity PID control. Each control gain was determined by trial and error.

## 3. Virtual Matrix Approach

### 3.1. Virtual Vessel and Matrix

In this study, the concept of a virtual leader vessel and an agent was introduced for the formation control of swarm operations. The virtual leader vessel is a virtual vessel that exists in a formation to maintain a rigid body, while agents are vessels that are located around the virtual leader vessel.

The key to the swarm operation is easy formation change and maintenance. For this purpose, a virtual matrix approach is proposed in this study. The virtual matrix is

generated as an $n \times m$ matrix, based on the location information of the virtual leader vessel. The size of the virtual matrix, the distance $d_{row}$ between rows, and the distance $d_{col}$ between columns can be set according to the formation. Further, the coordinates of each cell in the matrix are calculated according to the virtual leader vessel position, $d_{row}$ and $d_{col}$. Therefore, when the virtual matrix and the virtual leader vessel are determined, each agent is provided a cell number to be located on the virtual matrix according to the formation.

Figure 2 shows the virtual leader vessel and the $5 \times 5$ virtual matrix that is created. The virtual leader vessel is represented by a blue vessel to distinguish it from the agent. The $d_{row}$ and $d_{col}$ of the virtual matrix are set to 15 m each. Further, each agent is provided a cell number of the virtual matrix [(1,3), (3,2), (3,4), (5,1), (5,5)].

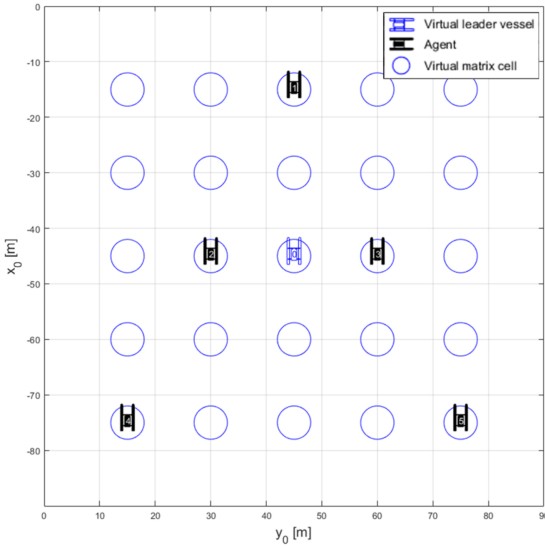

**Figure 2.** Wedge formation with virtual leader vessel, agents, and matrix.

The cases shown in Figure 3a–f represent the wedge, inverted wedge, T, inverted T, column, and row formations, respectively. Table 2 lists the virtual matrix cell number according to each formation. When the virtual leader vessel (with the virtual matrix attached) follows the global waypoint, which is the way point of the entire formation, the agent tracks the coordinates of a particular cell number that is provided to each agent. Therefore, the formation is maintained during operation. Through this process, the cell number designated to the agent changes if the formation changes, and agents move to change their location in the virtual matrix while the matrix works as a rigid body.

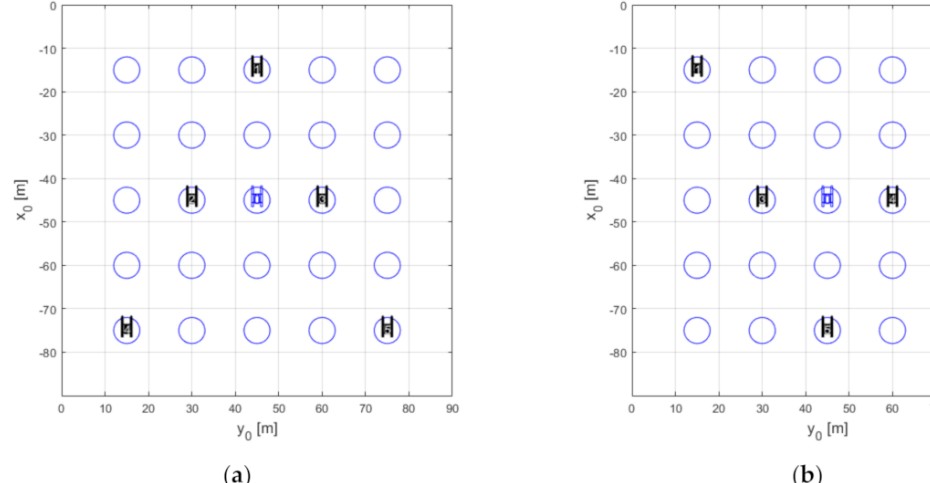

(**a**)    (**b**)

**Figure 3.** *Cont.*

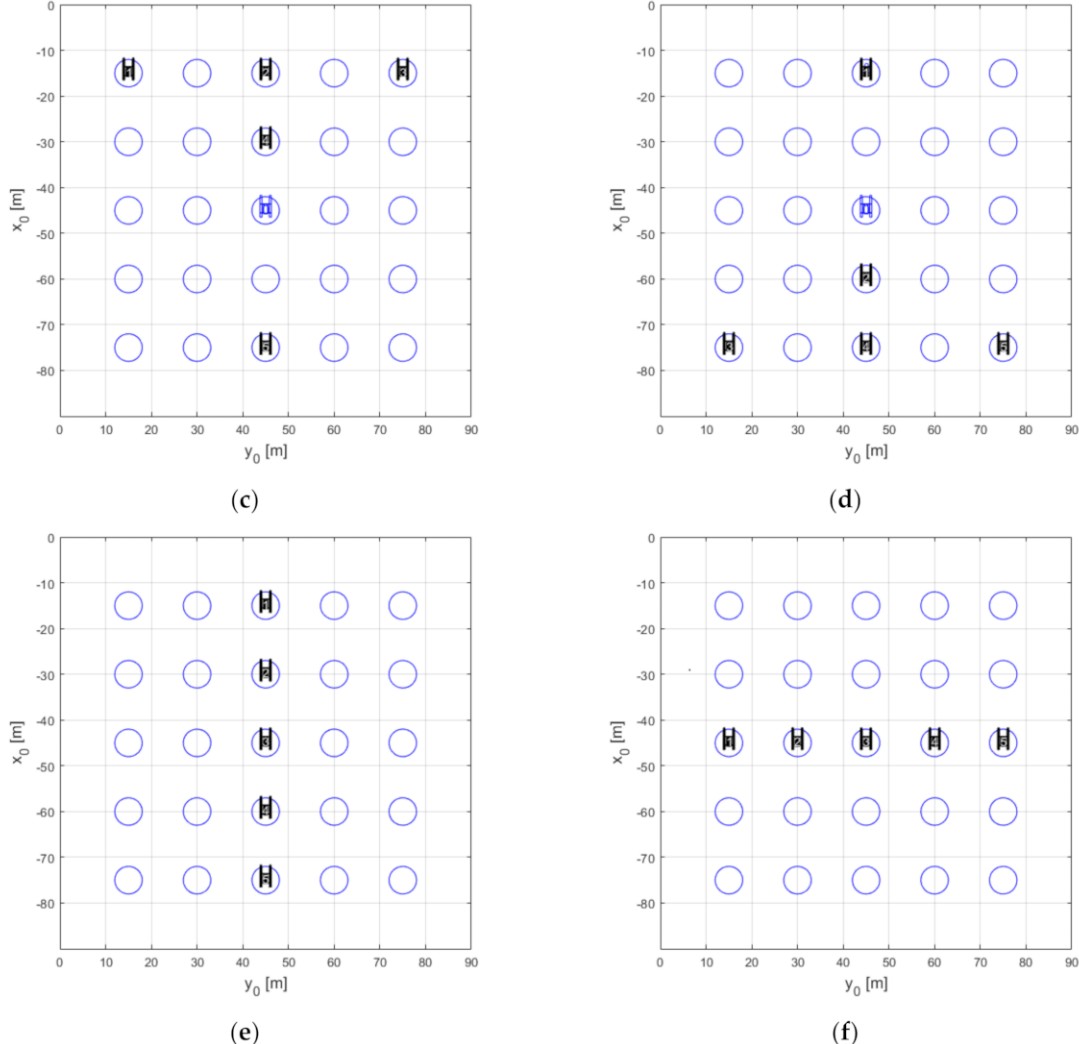

**Figure 3.** Different types of formations with a virtual leader vessel-located matrix center: (**a**) wedge formation; (**b**) inverted wedge formation; (**c**) T formation; (**d**) inverted T formation; (**e**) column formation; (**f**) row formation.

**Table 2.** Cell number for each formation.

| Formation | Cell |
|---|---|
| Wedge | [(1,3), (3,2), (3,4), (5,1), (5,5)] |
| Inverted wedge | [(1,1), (1,5), (3,2), (3,4), (5,3)] |
| T | [(1,1), (1,3), (1,5), (2,3), (5,3)] |
| Inverted T | [(1,3), (4,3), (5,1), (5,3), (5,5)] |
| Column | [(1,3), (2,3), (3,3), (4,3), (5,3)] |
| Row | [(3,1), (3,2), (3,3), (3,4), (3,5)] |

*3.2. Simulation for Formation Maintenance and Transformation*

A simulation was performed to verify the formation maintenance and transform performance of the swarm control applied with the virtual matrix approach. The simulation was performed under the assumption that no error exists in the position information of the virtual leader vessel, and the communication status between the virtual leader vessel and the agent is stable. In the simulation, the virtual leader vessel is numbered 0, and the agents in the formations thereafter are numbered in order starting at 1. The simulation was performed under the assumption that no error exists in the position information of

the virtual leader vessel, and the communication status between the virtual leader vessel and the agent is stable. The initial speed of each agent and virtual leader vessel is 0.8 m/s, the initial position of virtual leader vessel is (0,0), and the global waypoint that the virtual leader vessel follows is (400,0) in NED coordinates. The simulation runs for 500 s; during simulation, the virtual matrix size, $d_{row}$, and $d_{col}$ are 5 by 5, 15 m, and 15 m, respectively.

Figure 4 shows the simulation results to verify the formation-transform performance of swarm control with the virtual matrix approach. The formation-transform simulation comprises phase 1 forming the column formation, phase 2 forming the wedge formation, and phase 3 forming the row formation. Agents 1–5 were assigned different cell numbers according to the phase, as shown in Table 3. The virtual leader vessel (blue) is in cell (3,3) of the matrix. During a phase change, the agents follow the designated cell by the LOS guidance algorithm, while the heading angle and speed are controlled via PID. Accordingly, the formation is transformed and centered on the virtual leader vessel.

**Table 3.** Allocated cell number of each agent according to formation.

|         | Agent 1 | Agent 2 | Agent 3 | Agent 4 | Agent 5 |
|---------|---------|---------|---------|---------|---------|
| Phase 1 | (1,3)   | (2,3)   | (3,3)   | (4,3)   | (5,3)   |
| Phase 2 | (1,3)   | (3,2)   | (3,4)   | (5,1)   | (5,5)   |
| Phase 3 | (1,3)   | (1,2)   | (1,4)   | (1,1)   | (1,5)   |

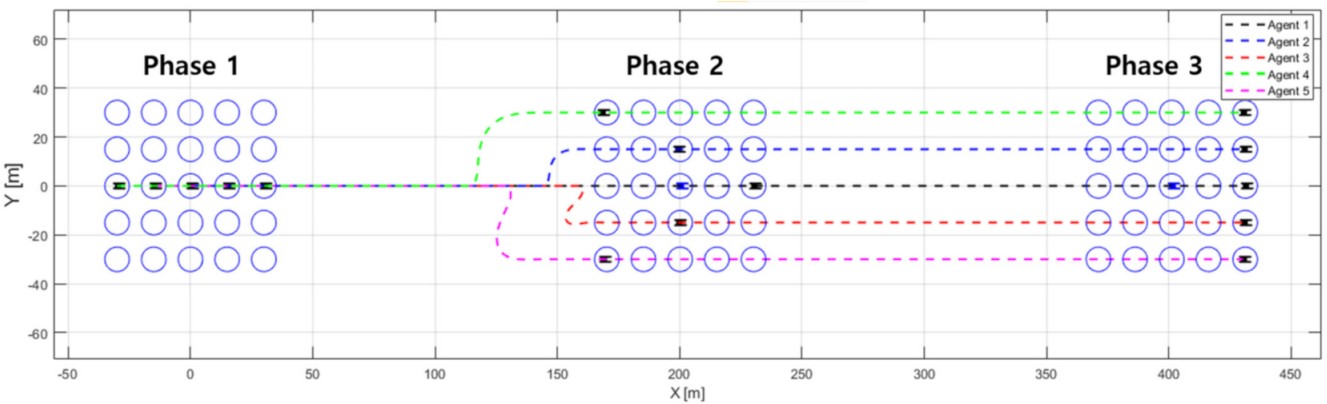

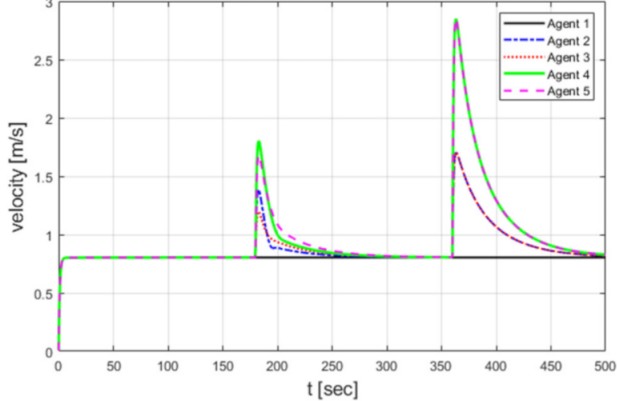
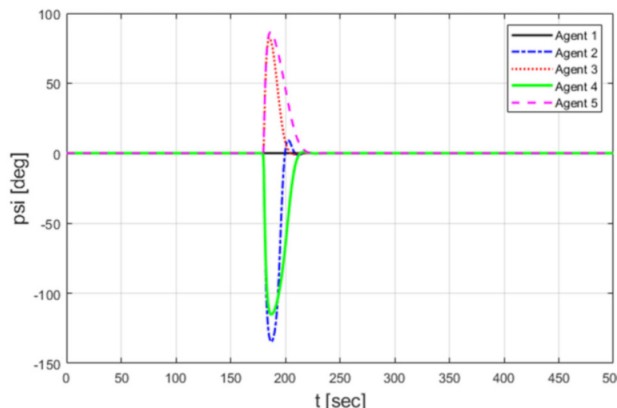

**Figure 4.** Formation maintenance and transform simulation.

### 3.3. Virtual Leader Vessel Location

Because the virtual matrix rotates according to the heading angle of the virtual leader vessel, the radius of gyration of the virtual matrix varies according to the virtual leader

vessel location. In this section, the case where the virtual leader vessel is located in the MC and GC is analyzed via a simulation that follows a zigzag-shaped global waypoint. When the virtual leader vessel is located at the MC, it is at the center of the matrix. In the case of GC, the virtual leader vessel is located on the average of the agent positions, and the coordinates of the virtual leader vessel ($x_{vl}$, $y_{vl}$) are calculated using Equation (17). Figure 5 shows the virtual leader vessel placed in the GC for each formation in Section 3.1.

$$x_{vl} = \sum_{i=1}^{n} \frac{x_i}{n}$$
$$y_{vl} = \sum_{i=1}^{n} \frac{y_i}{n} \text{ where } n = \text{agent number}$$

(17)

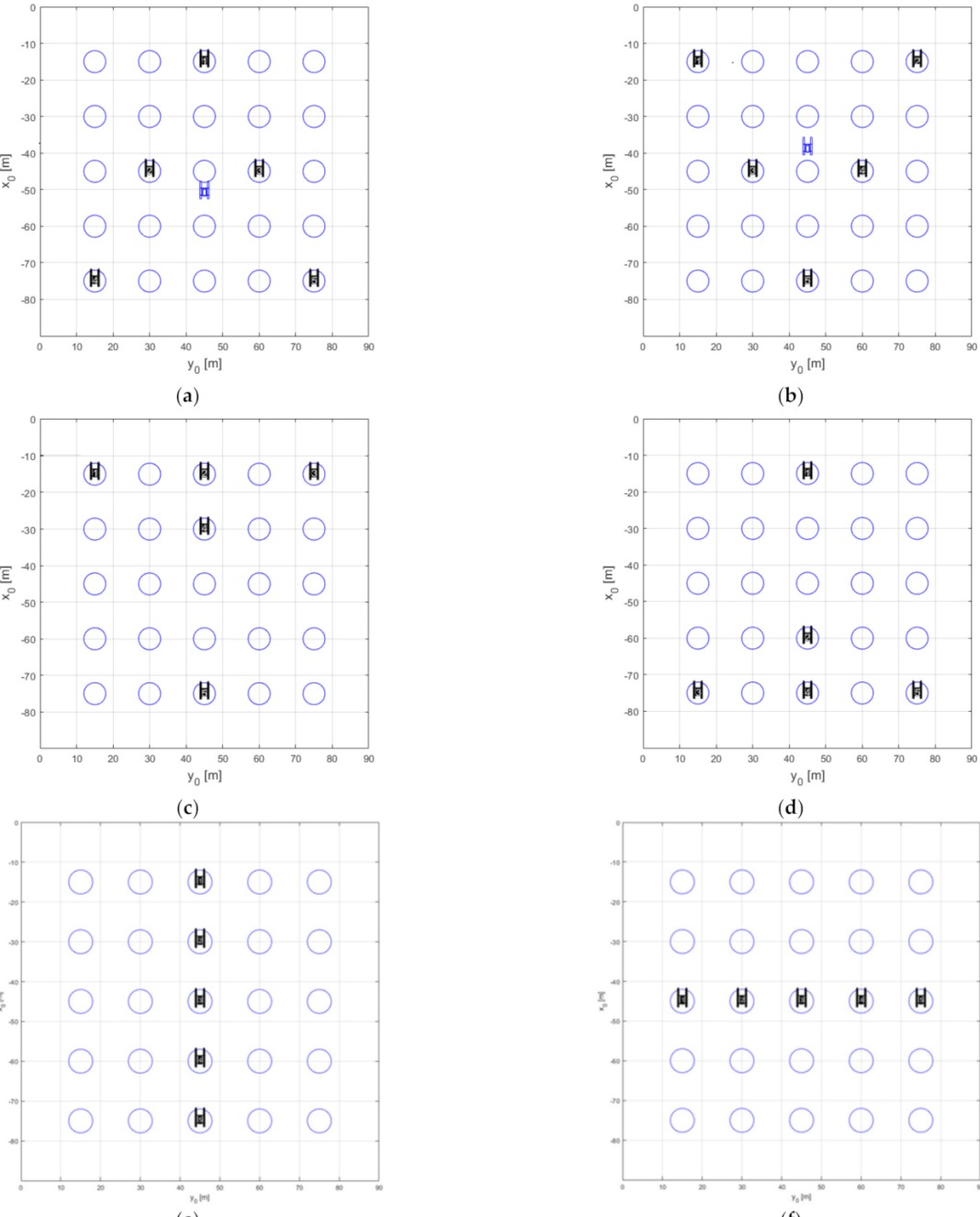

**Figure 5.** Different types of formations with a virtual leader vessel-located geometric center: (**a**) wedge formation; (**b**) inverted wedge formation; (**c**) T formation; (**d**) inverted T formation; (**e**) column formation; (**f**) row formation.

To compare the formation maintenance and efficiency of the swarm operation according to the two different virtual leader vessel locations, when the global waypoint of the virtual leader vessel is the same, the average of the trajectory distances of the agents, the average of the error distances between the agent and the given cell's coordinates, and the average of the battery usage are calculated. The average of the trajectory distances is calculated using Equation (18). Battery usage is calculated based on "the maximum speed that can be operated is approximately 5 knots, and when operated at 3 knots, it can be operated for approximately 3 h" [2] (pp. 67).

$$distance = \sum_{i=1}^{n} \frac{\int_0^t U_i dt}{n} \tag{18}$$

where $n$ is the agent number, $U$ is the velocity, and $t$ is the simulation time.

The results of each formation based on the two different virtual leader vessel locations are shown as a bar graph in Figure 6. The blue and red bars indicate the result when the virtual leader vessel is in the MC and GC, respectively. However, regardless of MC and GC, when the virtual leader vessel follows the same trajectory, a lower trajectory distance for the agent and low battery usage imply high efficiency, and a smaller error in the distance between the agents from the provided cell indicates high robustness of the formation. Most of the formations showed less trajectory distance, error distance, and battery usage when the virtual leader vessel was located in the GC. However, only wedge formations resulted in high efficiency and robustness when the virtual leader vessel was located in the MC.

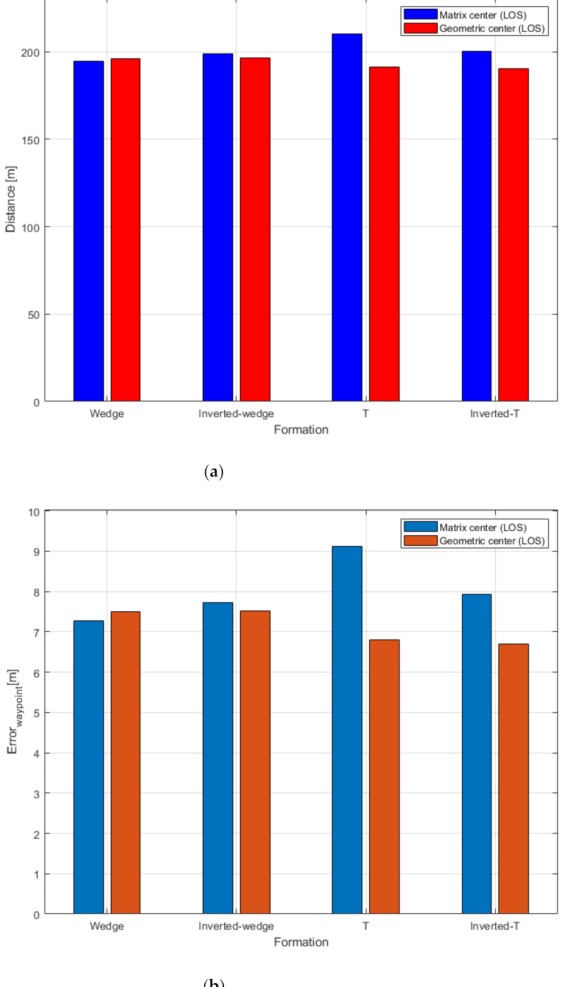

(a)

(b)

**Figure 6.** *Cont.*

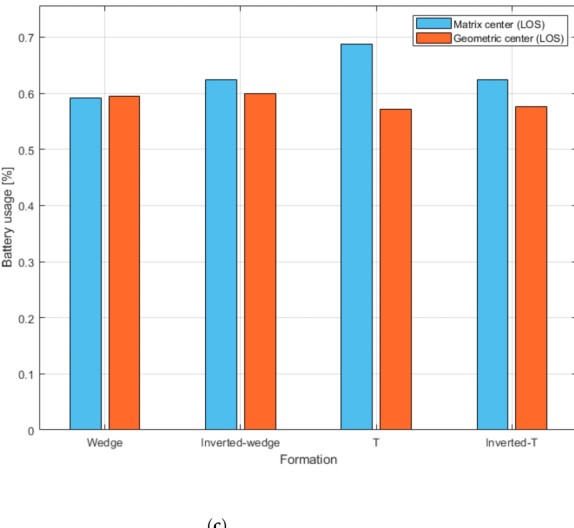

(**c**)

**Figure 6.** (**a**) Distance average of the agent according to formation. (**b**) Error distance average of the agent according to formation. (**c**) Battery usage average of the agent according to formation.

## 4. ISOT Guidance Algorithm

Each agent follows the designated cell using the LOS guidance algorithm and maintains its formation. However, when the virtual leader vessel rotates, the virtual matrix rotates, and certain agents show inefficient trajectories: 1. As shown in Figure 7, the green square area of the virtual matrix is rotated to the red square area when the virtual leader vessel is rotated. 2. Accordingly, the designated cell (5,5) of agent 2 in the green square exhibits the trajectory of the arc indicated by the blue arrow with a radius equal to the distance from the virtual leader vessel. 3. The agent, which follows the cell with a LOS guidance algorithm, moves backward of the leader vessel's forward direction. 4. This represents an inefficient movement, as indicated by the blue arrow in Figure 8, because the moving distance is longer than the path that the agent takes straight to the position where the agent should be located immediately after the rotation of the matrix is finished. This behavior series is defined as "go-back behavior."

The ISOT guidance algorithm was introduced to eliminate such inefficient maneuvers. As shown in Figure 9, the angle $\psi_v$ between the *x*-axis of the earth-centered fixed-coordinate system and the blue arrow is the heading angle of the virtual leader vessel and the rotation angle of the virtual matrix. Further, the angle $\psi$ between the *x*-axis of the earth-centered fixed-coordinate system and the red arrow is the heading angle of each agent. The angle $\psi_d$ between the *x*-axis of the earth-centered fixed-coordinate system and the green arrow is the desired heading angle of each agent calculated by the LOS guidance algorithm.

The $\text{LOS}_{\text{WP}}$ in Figure 10 is the location of the cell to be followed by the agent: $d_m$ is defined as the length between the agent's current position and $\text{LOS}_{\text{WP}}$, $\text{ISOT}_{\text{WP}}$ is defined as a point separated by $d_m$ from the $\text{LOS}_{\text{WP}}$ in the moving direction, and the distance between the agent and $\text{ISOT}_{\text{WP}}$ is $d_a$. According to the ISOT guidance algorithm, an isosceles triangle can be observed with the $\text{LOS}_{\text{WP}}$, the position of the agent, and the $\text{ISOT}_{\text{WP}}$. Accordingly, the desired heading angle of the agent is calculated using Equation (19). The agent follows the ISOT guidance algorithm when the difference between $\psi_v$ and $\psi_d$ is greater than 90°, and follows the LOS guidance algorithm when it is less than 90°.

$$\psi_{d,\ ISOT} = \begin{cases} \tan^{-1} \frac{(y_w - y_s)}{(x_w - x_s)}, & |\psi_v - \psi_d| < \frac{\pi}{2} \\ \tan^{-1} \frac{(y_w + d_m \sin(\psi_v) - y_s)}{(x_w + d_m \cos(\psi_v) - x_s)}, & |\psi_v - \psi_d| \geq \frac{\pi}{2} \end{cases} \tag{19}$$

The speed of the agent is controlled as shown in Equation (20). $U_m$ is the velocity of the virtual leader vessel and $\Delta t$ is the time required for the virtual leader vessel to move $d_m$.

$$U_a = \frac{d_a}{\triangle t} = \frac{d_a}{d_m} U_m \tag{20}$$

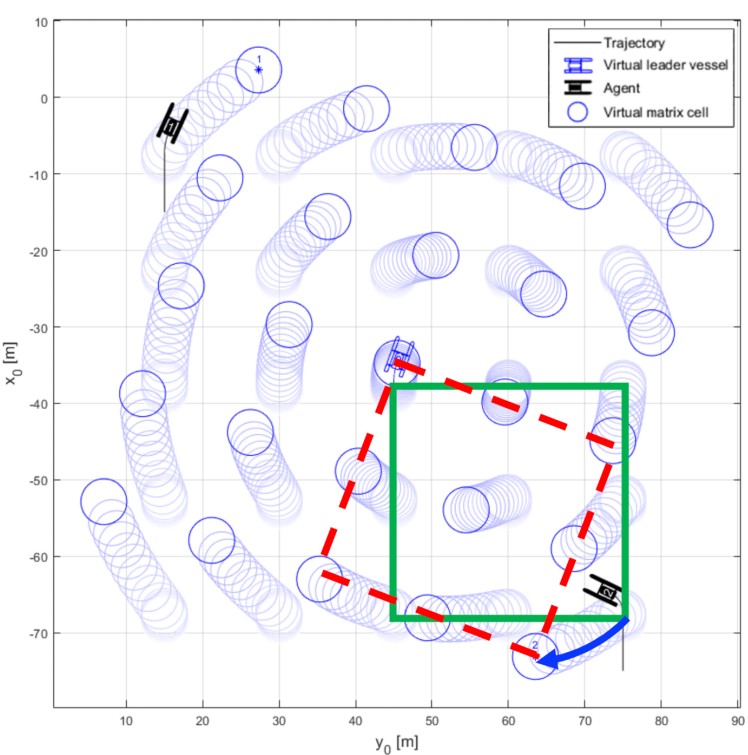

**Figure 7.** Matrix trajectory in a rotating situation.

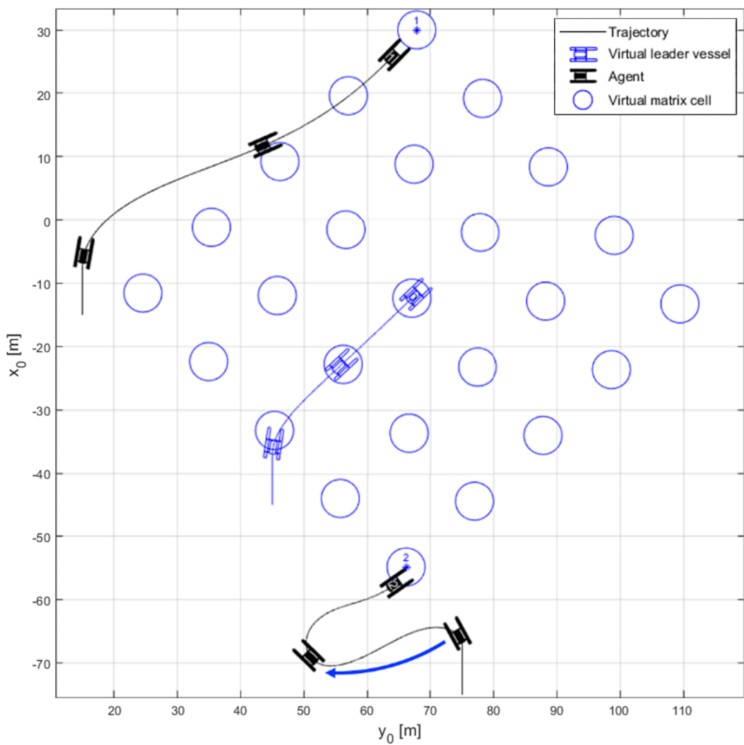

**Figure 8.** Go-back behavior.

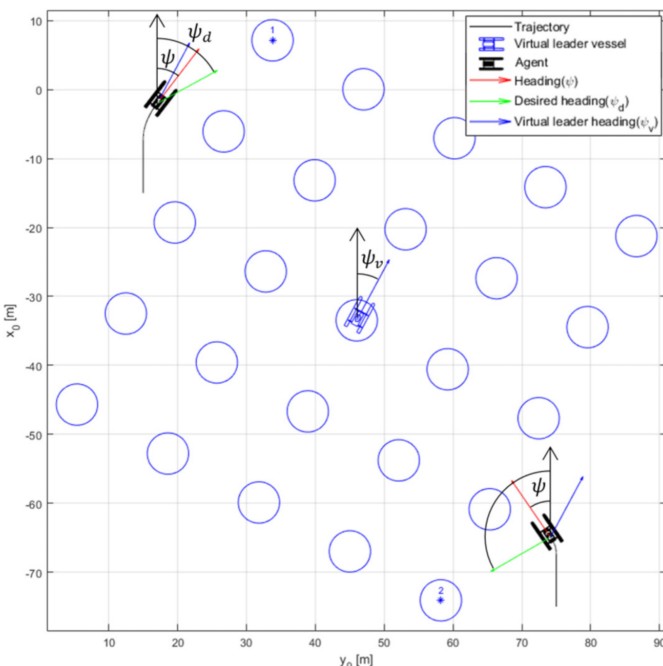

**Figure 9.** Indication of $\psi$, $\psi_d$, and $\psi_v$ for each vessel.

Figure 11 shows time trajectories of the LOS and ISOT algorithms. The agent following the LOS guidance is shown in red, and the trajectory is indicated by a red line, while the agent following the ISOT guidance is black, and the trajectory is indicated by a blue dotted line. At 20 s, the virtual leader vessel and virtual matrix rotate. If the agent follows the LOS guidance, the agent operates inefficiently because it follows the rotating virtual matrix. However, if the agent follows ISOT guidance, it operates efficiently by moving the minimum distance at the optimal speed. In the case of agent 1, because $|\psi_v - \psi_d|$ is within 90°, both LOS and ISOT generate the same waypoint.

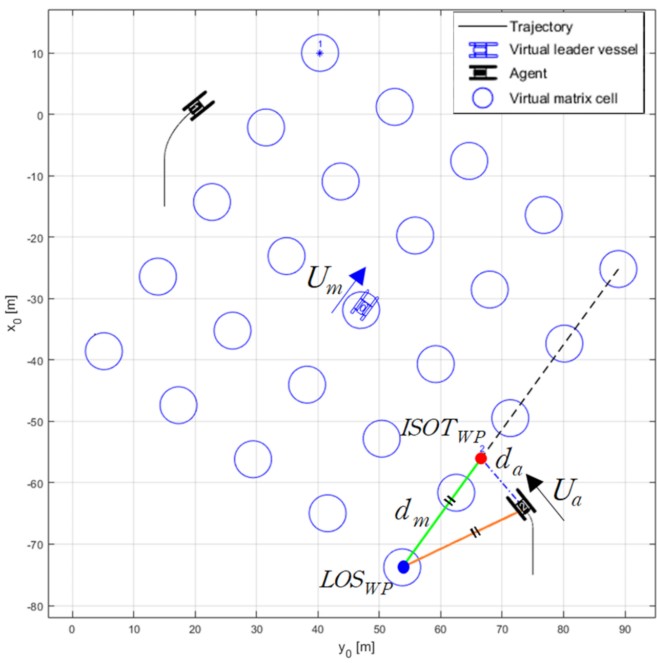

**Figure 10.** ISOT guidance algorithm description.

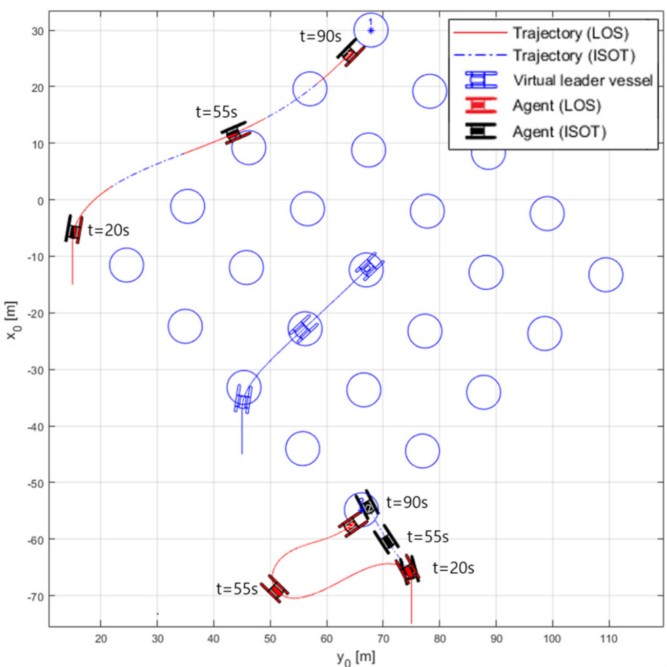

**Figure 11.** Trajectory comparison of LOS and ISOT guidance algorithms.

*Comparison of the ISOT and LOS Guidance Algorithms*

The combination of the algorithms and the derived virtual leader vessel position can be expressed by four cases: MC + LOS, LOS + GC, MC + ISOT, and ISOT + GC. Table 4 shows the results of analyzing these four cases by formation, as described in Section 3.3. When ISOT guidance was followed in all formations, both the MC and GC distance decreased compared to LOS guidance. It was confirmed that the average decreased by 14.24% and 13.68% for MC and GC, respectively. Further, battery usage also decreased by 21.66% and 20.59% for MC and GC, respectively, suggesting that the ISOT guidance algorithm is more efficient than LOS guidance. In contrast, the error up to the waypoint increased by 23.78% and 24.47% for MC and GC, respectively, confirming that the ISOT guidance robustness is lower than that of LOS guidance. Figure 12 shows the values in Table 4 as a bar graph. The first and second bars of each formation show the MC and GC cases based on the LOS guidance algorithm, whereas the third and fourth bars show the MC and GC cases based on the ISOT guidance algorithm.

Based on the results, the efficiency score is obtained by normalizing from 50 to −50 based on the largest and smallest values with the battery usage. In addition, the formation robustness score is obtained by normalizing based on the largest and smallest $Error_{waypoint}$. Figure 13 shows the scatter graph for each formation according to the combination of the guidance algorithm and the virtual leader vessel location. The *x*-axis represents the formation score and the *y*-axis represents the efficiency score. Red and blue markers indicate the results for GC and MC, respectively.

From the analysis of the guidance algorithm, it can be observed that the ISOT guidance algorithm in the red region is more efficient than the LOS guidance algorithm in the blue region. However, it can be observed that the LOS guidance algorithm has higher formation retention than the ISOT guidance algorithm. As a result of analyzing the virtual leader vessel position, MC is found to be more efficient than the GC of the wedge and inverted T formations in ISOT guidance. In contrast, in LOS guidance, GC is more efficient in all formations except the wedge formation.

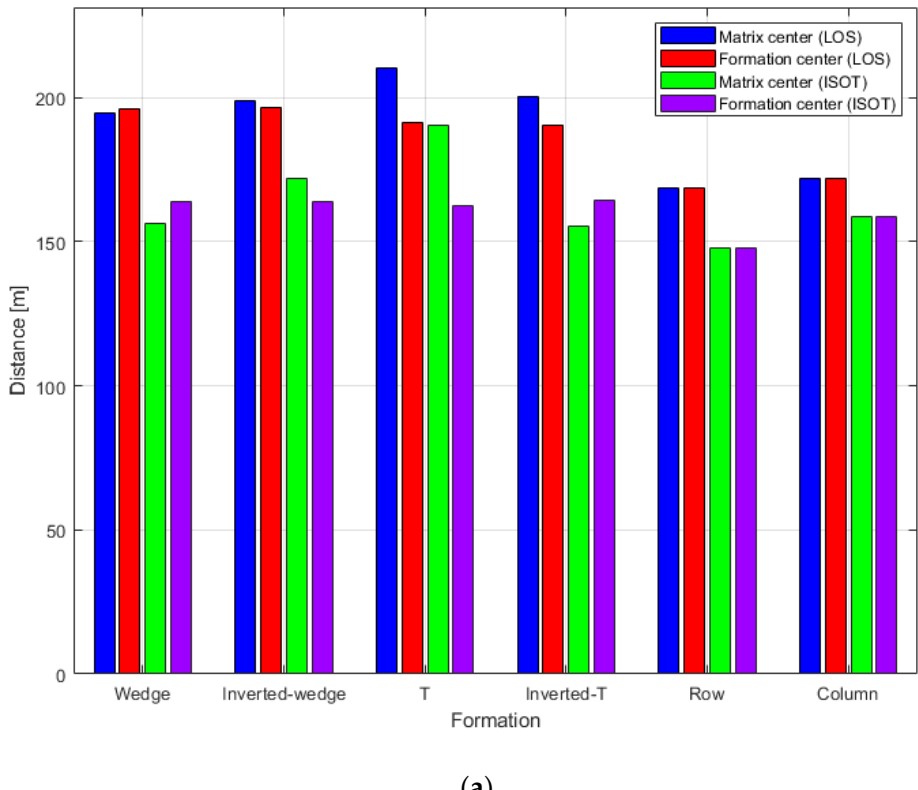

(**a**)

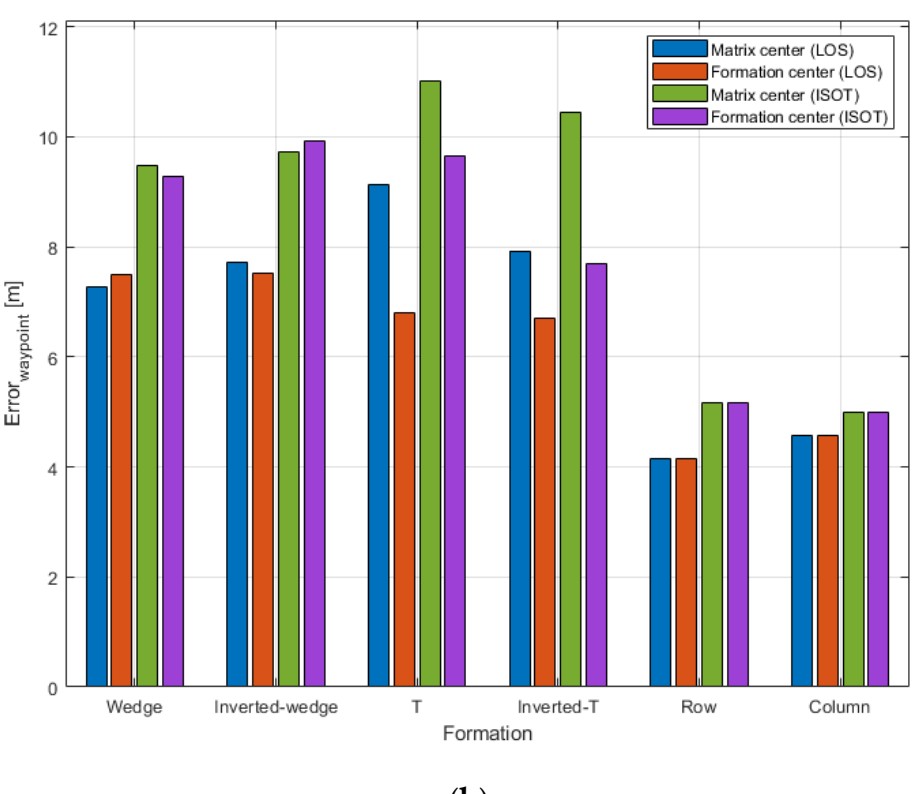

(**b**)

**Figure 12.** *Cont.*

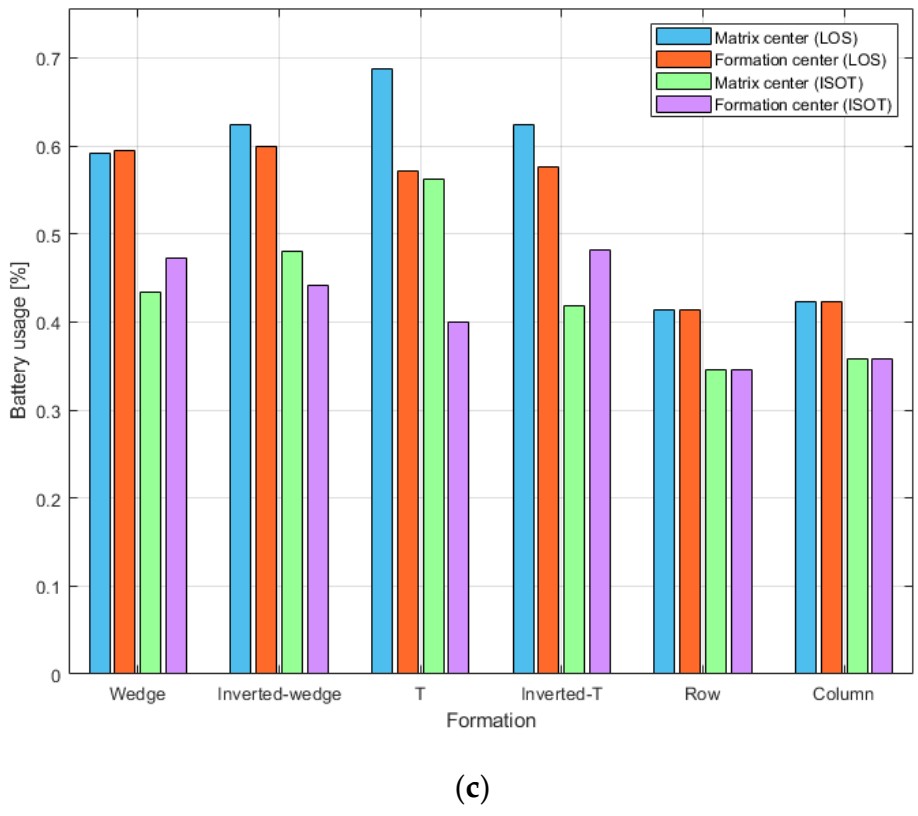

(**c**)

**Figure 12.** (**a**) Distance average of the agent according to formation using LOS, ISOT guidance. (**b**) Error waypoint average of the agent according to formation using LOS, ISOT guidance. (**c**) Battery usage average of the agent according to formation using LOS, ISOT guidance.

**Table 4.** Total result for guidance and location of the virtual leader vessel.

| Formation | | LOS | | ISOT | |
|---|---|---|---|---|---|
| | | Matrix Center (MC) | Geometric Center (GC) | Matrix Center (MC) | Geometric Center (GC) |
| Wedge | Distance [m] | 194.72 | 196.14 | 156.10 | 164.10 |
| | $Error_{waypoint}$ [m] | 7.27 | 7.50 | 9.48 | 9.29 |
| | Battery usage [%] | 0.592 | 0.594 | 0.43 | 0.47 |
| Inverted wedge | Distance [m] | 198.76 | 196.61 | 172.06 | 163.71 |
| | $Error_{waypoint}$ [m] | 7.73 | 7.53 | 9.72 | 9.92 |
| | Battery usage [%] | 0.62 | 0.60 | 0.48 | 0.44 |
| T | Distance [m] | 210.25 | 191.34 | 190.38 | 162.42 |
| | $Error_{waypoint}$ [m] | 9.12 | 6.80 | 11.01 | 9.66 |
| | Battery usage [%] | 0.69 | 0.57 | 0.56 | 0.40 |
| Inverted T | Distance [m] | 200.41 | 190.57 | 155.18 | 164.33 |
| | $Error_{waypoint}$ [m] | 7.93 | 6.69 | 10.45 | 7.70 |
| | Battery usage [%] | 0.62 | 0.58 | 0.42 | 0.48 |
| Column | Distance [m] | 168.46 | | 147.71 | |
| | $Error_{waypoint}$ [m] | 4.15 | | 5.18 | |
| | Battery usage [%] | 0.41 | | 0.35 | |
| Row | Distance [m] | 171.89 | | 158.45 | |
| | $Error_{waypoint}$ [m] | 4.56 | | 4.98 | |
| | Battery usage [%] | 0.42 | | 0.36 | |

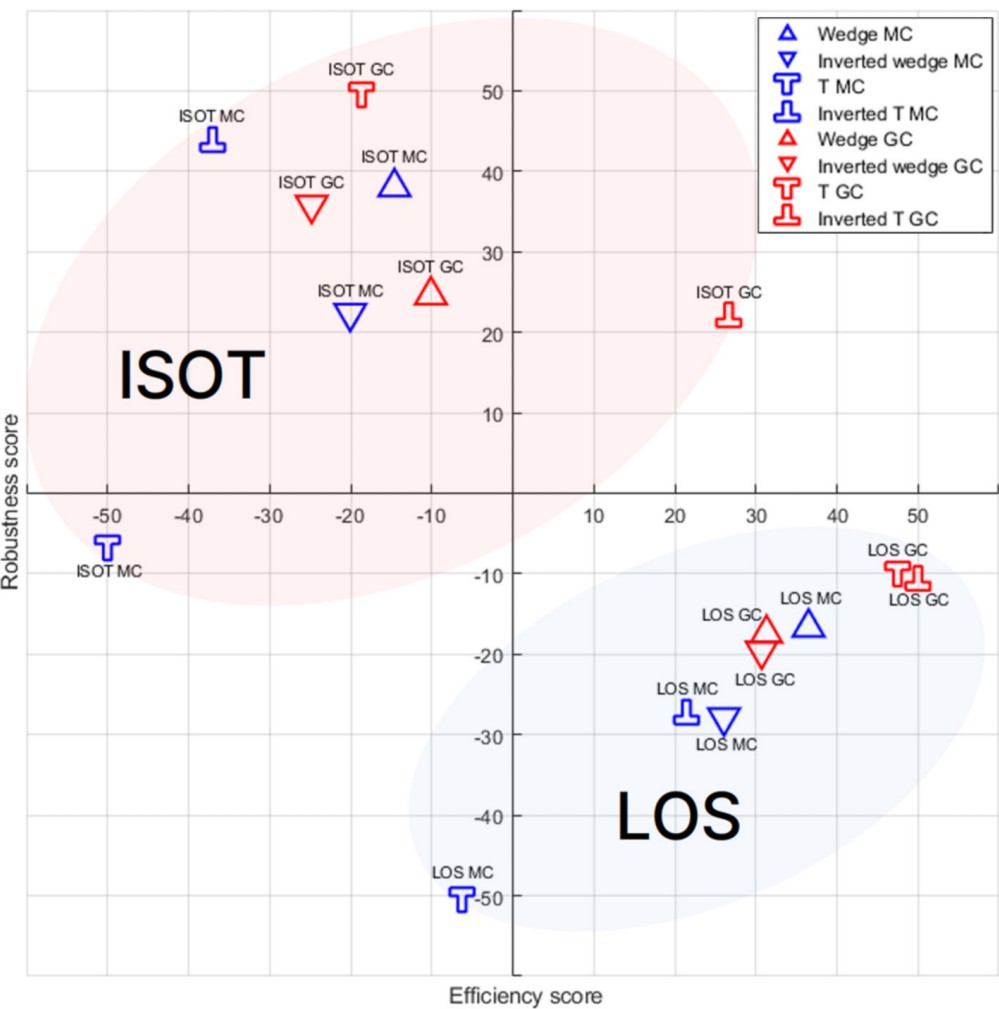

**Figure 13.** Comparison of the LOS and ISOT guidance algorithms in terms of efficiency and robustness score.

## 5. Swarm Simulation

Figure 14 shows a simulation that includes both changing the distance between the rows and columns of the virtual matrix and the formation transform of the swarm operation. An arbitrary map was created, and a swarm simulation was performed based on the map. The swarm simulation consisted of a total of eight phases, the total simulation time was 3300 s, and the ISOT guidance algorithm was applied with the virtual leader vessel located in the GC. The various phases were as follows: phase 1, row formation; phase 2, wedge formation; phase 3, narrow row formation; phase 4, column formation; phase 5, inverted T formation; phase 6, inverted wedge formation; phase 7, evasion formation to avoid islands; and phase 8, widened row formation. The distance between the columns of the virtual matrix decreased and increased in accordance with the width of the river by the specified value, indicating the convenience and user-friendly features of the virtual matrix approach. The simulation conditions for each phase are summarized in Table 5. Figure 15 shows a snapshot of each phase.

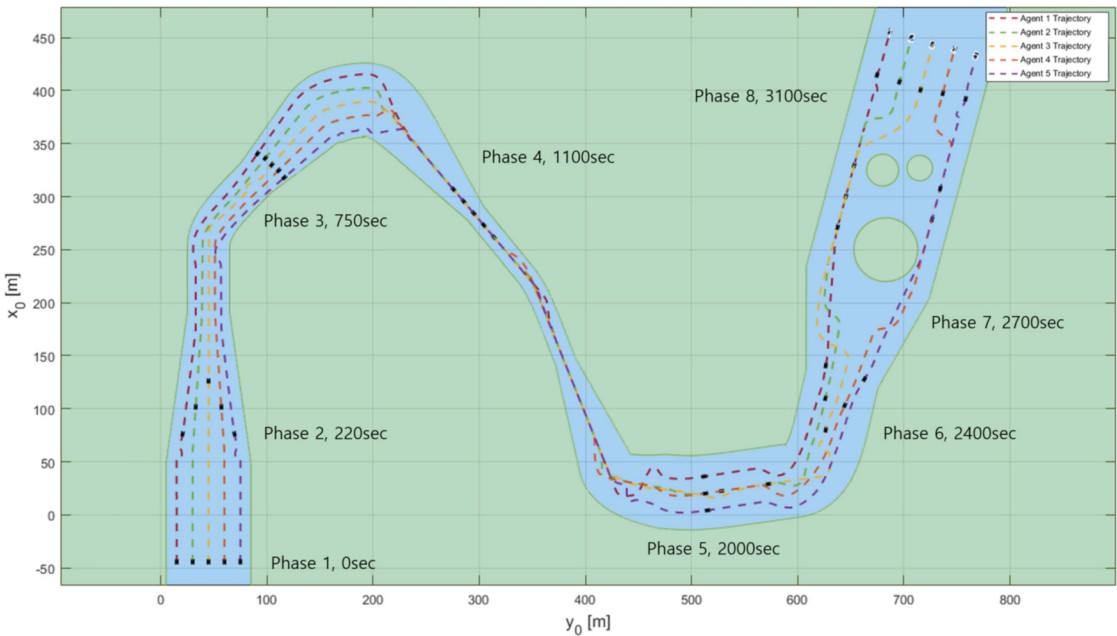

**Figure 14.** Trajectory of five agents in the swarm simulation.

**Table 5.** Condition of the swarm simulation according to phase.

|  | Time [s] | Guidance | Virtual Matrix Size | Virtual Leader Vessel Location | Formation | Distance between Rows [m] | Distance between Columns [m] |
|---|---|---|---|---|---|---|---|
| Phase 1 | 0~220 | ISOT | $5 \times 5$ | GC | Row | 15 | 15 |
| Phase 2 | 220~750 | ISOT | $5 \times 5$ | GC | Wedge | 8 | 6 |
| Phase 3 | 750~1100 | ISOT | $5 \times 5$ | GC | Row | 8 | 13 |
| Phase 4 | 1100~2000 | ISOT | $5 \times 5$ | GC | Column | 15 | 4 |
| Phase 5 | 2000~2400 | ISOT | $5 \times 5$ | GC | Inverted T | 15 | 8 |
| Phase 6 | 2400~2700 | ISOT | $5 \times 5$ | GC | Inverted Wedge | 15 | 8 |
| Phase 7 | 2700~3100 | ISOT | $5 \times 5$ | GC | Avoidance | 15 | 23 |
| Phase 8 | 3100~3300 | ISOT | $5 \times 5$ | GC | Row | 15 | 23 |

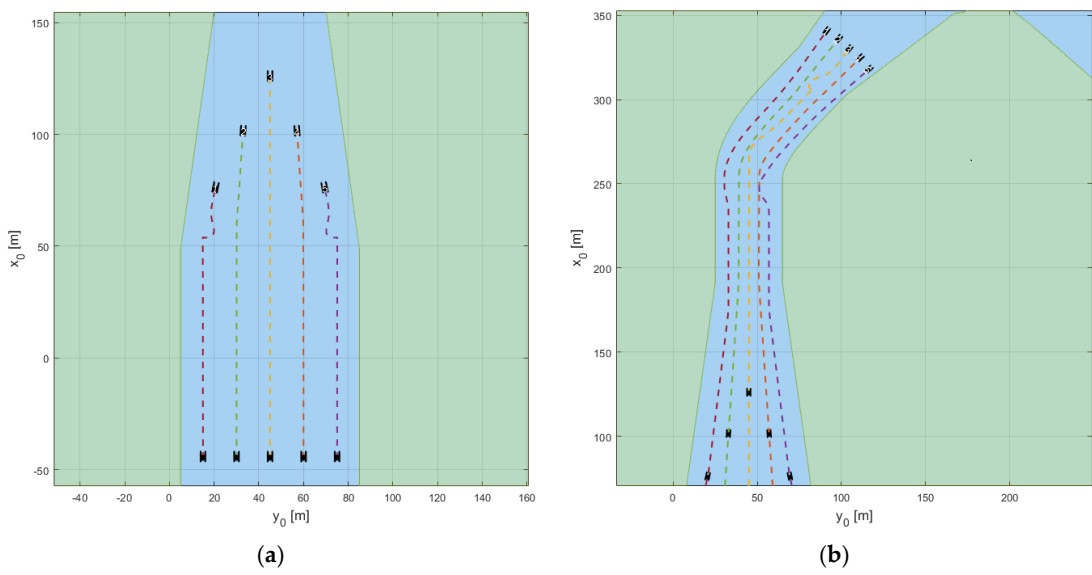

(**a**)　　　　　　　　　　　　　　　　　　　　　(**b**)

**Figure 15.** *Cont.*

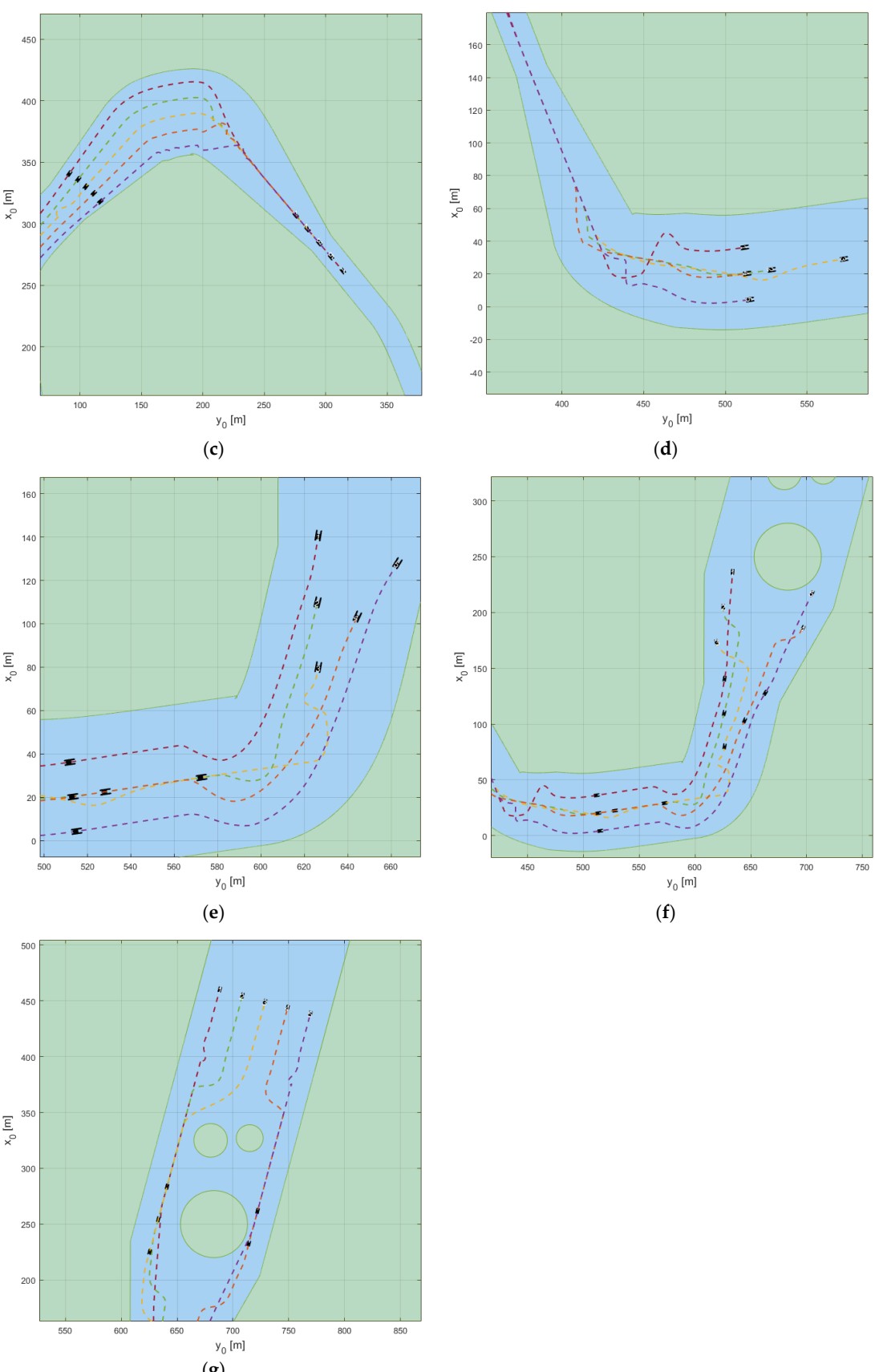

**Figure 15.** Snapshot of (**a**) phase 1; (**b**) phase 2; (**c**) phase 3 and 4; (**d**) phase 5; (**e**) phase 6; (**f**) phase 7; (**g**) phase 8.

## 6. Conclusions

In this study, the concept of a virtual leader vessel and agent was introduced for the formation control of multiple ASVs. The virtual matrix approach was suggested to generate a virtual rigid matrix, which is calculated based on a virtual leader vessel, to improve the robustness and scalability of the formation. This approach is a user-friendly formation control algorithm wherein the user only needs to specify the cell number, which facilitates control in practice. Simulations applied with the proposed approach were performed to confirm the formation maintenance and transformation performance of the entire formation when the virtual leader vessel travels straight and turns. Furthermore, in this simulation, the maneuvers of each agent depend on the virtual leader vessel location; hence, the efficiency and robustness of the formation according to the MC and GC, which are the virtual leader vessel locations, were analyzed.

Regardless of the virtual leader vessel position, the robustness of the overall formation that follows the LOS guidance algorithm was high; however, each agent showed inefficient trajectories when the matrix rotated. Therefore, an ISOT guidance algorithm was proposed. With this guidance, the agent's "go-back behavior" was improved and the overall efficiency enhanced, resulting in a decreased battery usage of 21.66% for MC and 20.59% for GC. In addition, the formation maintenance and transformation performance of the proposed algorithm were verified by performing a search simulation using a virtual matrix approach and the ISOT guidance algorithm.

Despite its many advantages, this study had several limitations. First, because a single dynamic model was applied to each agent, environmental factors between agents were not considered. Second, because a simulation-based swarm operation was performed, a practical issue may occur when the algorithm is applied to real environments. Therefore, in future work, in the case of unstable communication or a problem with the virtual leader vessel, the algorithm should be improved such that the agent can self-operate according to the situation. In addition, a model ship-based swarm system must be built, and swarm operations tests must be conducted in a real environment for comparison with simulation results.

**Author Contributions:** Writing—original draft preparation: S.-R.K.; investigation: H.-J.J.; formal analysis: J.-H.K.; supervision: J.-Y.P. All authors have read and agreed to the published version of the manuscript.

**Funding:** This work was supported by a grant from Brain Korea 21 Program for Leading Universities and Students (BK21 FOUR) MADEC Marine Designeering Education Research Group and "Development of Autonomous Ship Technology (20200615)" funded by the Ministry of Oceans and Fisheries (MOF, Korea).

**Institutional Review Board Statement:** Not applicable.

**Informed Consent Statement:** Not applicable.

**Conflicts of Interest:** The authors declare no conflict of interest.

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
