# Peer review of "Formation Control of Swarming Vessels Using a Virtual Matrix Approach and ISOT Guidance Algorithm"

_processes, doi:10.3390/pr9091581_

Round 1
Reviewer 1 Report
The authors prepared an interesting manuscript on formation control of multiple ASVs. Simulations applied with the proposed approach were performed to 334 confirm the formation maintenance. Overall, the presented results are detailed and reliable. However, in the current form, the manuscript needs minor revisions.
First, more interpretation on the figures and results are needed. The authors should mention the important findings and come up with some useful conclusions. More effort is encouraged.
The Conclusions section requires substantial rewriting in order to highlight all the significant findings in the manuscript. In the current form, the first paragraph is a summary of the work, and the last paragraph is a job outlook.
Some unclear sentences or words should be improved, eg. P4Line129:“(PID) () control”
Author Response
Please see the attachment. Note that I have added a list of abbreviations which was not included in the previous version of the manuscript.
Again, thank you for giving us the opportunity to strengthen our manuscript with your valuable comments and queries. We have worked hard to incorporate your feedback and hope that these revisions persuade you to accept our submission.
I would be happy to make any further changes that may be required.

Reviewer 2 Report
The subject of the research is very interesting, and the presented method is uncomplicated and promising in practical applications. Formation control, based on the virtual leader presented in this publication is an issue that is often discussed in robotic control and is also applicable to ASVs. Nevertheless, I suggest paying special attention to the following elements of the publication.
1. Abstract needs some corrections:
- In line 13 there is an abuse of 'the';
- Line 13 contains an incomprehensible phrase 'is defined as two cases';
- In line 17 occurs word 'Second', but in the earlier part it does not appear first;
- Why 'Isoscales' in line 17 starts with the capital letter?;
- In line 18 'inefficient behavior of certain agents' was used. The above wording is very general and needs clarification in order to encourage the reader to read ths publication.
2. Introduction gives insufficient research background. I suggest more deep references analysis in these fields:
- Virtual leader-follower control, which I think is the basis of this research. There are mane newer and intersting publications in this field, than these mentioned in Your Introduction, e.g. E. Abbasi, M. Ghayour and M. Danesh, "Virtual Leader-Follower Formation Control of Multi Quadrotors by using Feedback Linearization Controller," 2017 5th RSI International Conference on Robotics and Mechatronics (ICRoM), 2017, pp. 614-619, doi: 10.1109/ICRoM.2017.8466165.
- Ongoing ASV projects: In line 28 there is a reference to 'research in recent years', but these references are nto the newest publications in ASVs field. Moreover there are some ongoing research projects like: NFAS, AUTOSHIP, YARA Birkeland vessel.
- Formation control: There are some new publications concernig formation control, like: Cheng Liu, Qizhi Hu, Xuegang Wang, Distributed guidance-based formation control of marine vehicles under switching topology,
Applied Ocean Research or Zhang, G., Yu, W., Zhang, W. et al. Robust adaptive formation control of underactuated surface vehicles with the desired-heading amendment . J Mar Sci Technol (2021).
3. In the Introduction section in line 84 there is a phase 'ehxibited inefficient behavior', which is very general one. I suggest listing specific behaviors and naming them.
4. In line 112 equation should be replaced by Equation and in line 129 '()' is unnecessary.
5. In Figure 2 I suggest introduction of virtual leader and agent symbols distinguishable on the grayscale printout.
6. In my opinion in the figures axis descriptions and numbers are too small and illegible. I suggest using bigger font.
7. In line 161 I suggest not to start sentence with '(a)'. Please take into accoun starting it like that: Cases (a), (b)... or Figures 3(a), (b),... respectively represent...
8. Virtual matrix approach description is in my opinion incomplete.
- Starting from lines 166-167 it needs clarification in order to allow your results to be reproduced in other cases. ' the cell provided to the agent is changed in a situation where the formation is changed' requires an explanation of how this change is done.
- During ASV formation control all matrix moves. But there is a lack of description how particular agents and virtual leader reference positions for PID control are determined in the successive moments of time.
9. In subsection 3.2 simulation setup is described in a very limited way. Based on this description, it is not possible to reproduce the test results. Test case lacks a description of the method of determining the reference position and control law formulation. Moreover, in line 182 speed control is mentioned. But in my opinion transformation from row to column formation may not be done only via agents speed change. Agents need to change their course and speed in order to perform this maneouver. I suggest supplementing the simulation results with course and speed graphs for individual agents.
10. There is also a question if the possibility of a collision between agents is being monitored during the execution of a maneuver and what are the ways of avoiding it?
11. In line 197 'are' should be used instead of'is'.
12. You used 'the average of the trajectory distances of the agents' to compare the formation maintenance and efficiency. But in my opinion it should be defined how it is calculated and what level is the reference level for the interpretation of the results. In Figures 6(a) an 11(a) it is not known what should be taken as reference. Is it expected that results for MC will be the same as for GC? Or the average distance should be minimized or reach a predefined number?
13. What does 'inefficient trajectories' (in line 230) exactly men? I suggest defining them, naming and listing cases when these 'inefficient trajectories' occur.
14, In figure 6(a), (b) and (c) I suggest color unification. The same remark refers to the Figure 11. Descriptions in legend are the same for (a), (b) and (c) and bar colors are different for each one.
15. In Figure 12 there are two separate areas marked by red and blue, but they are not described in the text body and in figure legend. What they are used for?
16. In lines 307 and 324 there is a mistake in Figure numbering. There is Figure 3 instead od Figure 12 and Figure 43 instead of Figure 13.
17. In Figure 14 axis captions are too small and illegible.
18. Conclusions should be rewritten. They are very general. Imprecise terms such as high, improved etc. werre used.I suggest to introduce numerical indicators for efficiency and robustness, which will allow for a clear assessment of how the presented method allows for their improvement.
19. In References section, point 1 and 3 there are unnecessary dashes before name abbreviations (e.g. '-Y.').
Author Response

(The authors gave the same response as above.)
